# The Ninhydrin Reaction Revisited: Optimisation and Application for Quantification of Free Amino Acids

**DOI:** 10.3390/molecules29143262

**Published:** 2024-07-10

**Authors:** Amelie Charlotte Stauß, Carolin Fuchs, Paulina Jansen, Sarah Repert, Kimberley Alcock, Sandra Ludewig, Wilfried Rozhon

**Affiliations:** Department of Agriculture, Ecotrophology, and Landscape Development, Anhalt University of Applied Sciences, 06406 Bernburg, Saxony-Anhalt, Germany; ameliest@t-online.de (A.C.S.); carolin.fuchs@student.hs-anhalt.de (C.F.); paulinajansen99@gmail.com (P.J.); sarah.repert@hs-anhalt.de (S.R.); kimberley.alcock@student.hs-anhalt.de (K.A.); sandra.ludewig@hs-anhalt.de (S.L.)

**Keywords:** foods, free amino acids, hydrindantin, ninhydrin, spectrophotometry

## Abstract

The ninhydrin reaction is commonly used for the detection of amino acids. However, in the literature, different conditions with respect to the buffer system, its pH and concentration, type of organic solvent, incubation time, and temperature, as well as the concentrations of the reagents, are described. To identify the most suitable conditions, colour development with reagents of varying compositions and different reaction temperatures and times were investigated using asparagine as a model amino acid. Asparagine was selected since it is one of the most abundant free amino acids in many types of samples. The optimal reaction mixture consisted of 0.8 mol L^−1^ potassium acetate, 1.6 mol L^−1^ acetic acid, 20 mg mL^−1^ ninhydrin and 0.8 mg mL^−1^ hydrindantin in DMSO/acetate buffer 40/60 (*v*/*v*) (final concentrations). The best reaction condition was heating the samples in 1.5 mL reaction tubes to 90 °C for 45 min. Afterwards, the samples were diluted with 2-propanol/water 50/50 (*v*/*v*) and the absorbance was measured at 570 nm. The proteinogenic amino acids showed a similar response except for cysteine and proline. The method was highly sensitive and showed excellent linearity as well as intra-day and inter-day reproducibility.

## 1. Introduction

Free amino acids are crucial compounds in many foods and are frequently considered as a main factor in food quality. In tomato fruits, the total amount of free amino acids increases significantly during ripening on the vine [1]. Such an increase is not seen in tomatoes that have ripened off the vine [2], which may account for their different taste [3]. Similarly, in grape [4], melon (*Cucumis melo*) [5] and sunberry (*Physalis minima*) [6], the level of free amino acids increases during ripening while it remains unaffected in jujube fruits (*Ziziphus jujuba*) [7] and decreases in apple [8] and strawberry [9]. Wheat grown under sulphur starvation contains high amounts of free amino acids in the grains. Pastries made out of such wheat flowers have a reduced loaf volume, a dark crust and crumb, and contain significant amounts of the neurotoxin acrylamide [10]. The content of free amino acids is also a critical factor for sugar beet quality, where this parameter is referred to as α-amino nitrogen. Free amino acids are one of the most important factors preventing sugar crystallisation and thereby reduce yield. A low content is therefore mandatory for the high quality of sugar beets [11,12].

Free amino acids also play a fundamental role for the taste of many foods prepared via fermentation. For instance, the content of soluble nitrogen compounds, particularly free amino acids, is a major factor of soy sauce quality [13]. During cheese production, the content of free amino acids increases from approximately 50–200 mg/100 g to more than 1000 mg/100 g [14,15], which has been suggested as a good indicator of cheese ripening [16]. Therefore, the quantification of the total amino acid level can be informative for a wide variety of samples.

A number of techniques have been used to determine free amino acids in food and other sample types. Free amino acids are often analysed using so-called amino acid analysers [17,18,19,20]. Although reliable, disadvantages include low sensitivity and a long run time of typically more than 30 min for one sample. Frequently used alternatives are based on high-performance liquid chromatography in the reversed-phase (RP) or hydrophilic interaction (HILIC) mode with a precolumn derivatisation step to enhance the binding of the amino acids to the stationary phase and improve their detection sensitivity. Commonly used derivatisation reagents include phenyl isothiocyanate (PITC) [21,22,23,24], o-phthaldialdehyde (OPA) alone [25,26,27,28,29] or in combination with fluorenylmethyloxycarbonyl chloride (FMOC-Cl) [30], 6-aminoquinoline-*N*-hydroxy-succinimidyl carbamate (AQC) [31,32], dansyl chloride [33,34,35] and Sanger’s reagent (1-fluoro-2,4-dinitrophenylbenzene, DNFB) [36,37]. These methods are highly sensitive and allow the quantification of individual amino acids. However, they are time consuming, and require special equipment and a high degree of standardisation to allow the reliable assignment of all peaks.

Alternatively, spectroscopic methods can be used for the quantification of free amino acids. Using OPA, a highly sensitive method for the quantification of the free amino acids using spectrofluorimetry has been developed [38]. This method is simple and has superior sensitivity. Disadvantages include the instability of the formed isoindoles and the moderately uniform response of the different proteinogenic amino acids. Another well-known reagent is 2,4,6-trinitrobenzene sulphonate (TNBS) which gives, depending on the reaction conditions, an orange coloured reaction product with an absorption maximum at 420 nm [39] or a derivative with an absorption maximum at 340 nm [40]. The latter has the disadvantage that the absorption maximum overlaps with that of picric acid, a side product of the reaction [41], while the former originates from an unstable complex. Similarly, amino acids can be derivatised with DNFB under alkaline conditions to stable 2,4-dinitrophenyl derivatives, which can be extracted using ethyl acetate and measured at 420 nm [42]. However, DNFB hydrolyses rapidly under alkaline conditions to 2,4-dinitrophenol, which absorbs in the same range as the 2,4-dinitrophenyl amino acid derivatives [43]. Thus, it remains to be investigated how efficient extraction with ethyl acetate can discriminate between the amino acid derivatives and the by-product.

The most widely used reagent for the spectrophotometric quantification of amino acids is ninhydrin. This reagent reacts with primary amino groups to form Ruhemann’s purple, a colourful compound that can be measured using spectrophotometry at 570 nm. In contrast to the other spectroscopic methods described above, all primary amino compounds give the same dye because the final reaction product contains only the primary amino nitrogen of the amino acid. Thus, in theory, all primary amino compounds should give the same response. The ninhydrin reaction has been used for a long time, but there are several variations in its implementation. Typical reagents consist of ninhydrin dissolved in a mixture of an aqueous buffer and an organic solvent including 2-methoxyethanol, phenol, sulfolane and dimethyl sulfoxide (DMSO). As a buffer system, acetic acid/lithium acetate with a molar ratio of 0.325/1 is frequently used [44]. Interestingly, Sun et al. showed that lithium acetate has no advantage over the significantly cheaper sodium acetate [45]. Surprisingly, they provided evidence that the reaction works very well in the range of pH 8–9. In addition, an agent must be added that reduces a small quantity of ninhydrin to hydrindantin, which is required for an efficient reaction. All proposed compounds have disadvantages: tin(II)-chloride [46] forms insoluble precipitates, cyanide [47] and sodium borohydride [48] are highly toxic and ascorbic acid [49] increases the blank significantly. Instead of a reducing agent, pure hydrindantin can be added [44]. This has the advantages that it is not toxic and the blank is kept low. Finally, the reaction must be heated to induce the formation of Ruhemann’s purple. A wide range of different temperatures and incubation times have been reported. The most frequently used temperature was 100 °C for 10 to 30 min [45,50], but also lower temperatures including 90 °C for 15 min [51], 80 °C for 30 min [52] and even a temperature as low as 60 °C for 30 min [53] have been reported.

Although the ninhydrin reaction has been used for decades, a variety of reaction conditions are used and no standard procedure has been established. Many of the protocols used seem to be based on tradition rather than experimental evidence. Thus, we decided to investigate the factors affecting the formation of Ruhemann’s purple in a systematic way. Based on the literature, an important factor to be investigated is the type and concentration of the acetate buffer, particularly the cation of the salt and the acetic acid to acetate ratio. Among the proposed solvents, only sulfolane and DMSO are a good choice with respect to work safety and thus we focused on those. All previously used reducing agents have disadvantages and thus we decided to use hydrindantin instead because it is commercially available, simple to use and non-toxic. Furthermore, we investigated the impact of the ninhydrin and hydrindantin concentrations and the reaction conditions, particularly temperature and reaction time on dye formation.

## 2. Results and Discussion

### 2.1. Buffer Composition

In previous publications, acetic acid in combination with lithium or sodium acetate was used as a buffer system but no reasons for the selection of these cations were given [44,46,47,48,49,54]. To investigate the impact of the cation, reactions in the presence of lithium, sodium, potassium, magnesium, calcium and barium acetate were performed. Since preliminary experiments had indicated that a ratio of acetic acid to acetate of 2/1 (mol/mol) is best (Appendix A), this ratio was used for all acetates tested. Asparagine was used as a representative amino acid during the optimisation procedure since it is one of the most abundant free amino acids in many sample types, including food and plant extracts. As indicated in Figure 1A, the type of acetate had only a minor impact, since the absorption measured for all reactions containing asparagine was very similar. The same also applied to the blank reactions. The ratio of the asparagine reaction and the blank was highest for the three alkaline metal acetates and for magnesium, indicating that these acetates are equally suitable. Among them, potassium acetate has by far the highest solubility with 25 mol salt per kg water [55], allowing the preparation of buffers with a high capacity. In addition, it is one of the cheapest acetates. Thus, we selected potassium acetate for the further experiments.

As stated above, preliminary experiments showed that an acetic acid to lithium acetate ratio of 2/1 gave the best results. To test whether this also applies for the acetic acid/potassium acetate buffer system, the concentration of potassium acetate was kept at 0.5 mol L^−1^ while the concentration of acetic acid varied between 0 and 1.8 mol L^−1^. As for lithium acetate, a broad optimum in the range of approximately 0.7 to 1.4 mol L^−1^ acetic acid was observed. This corresponds to a ratio of acetic acid to potassium acetate of 1.4/1 to 2.8/1. This result (Figure 1B) was surprising, since in most previous publications buffer systems with a significantly lower ratio of acetic acid to acetate, typically in the range of 0–0.96/1, were used [45,47,54]. However, our data with acetic acid/potassium acetate (Figure 1B) and acetic acid/lithium acetate (Appendix A) clearly indicate that an acid to salt ratio of approximately 2/1 is most suitable, which corresponds to the results of Fisher et al. [47]. It is worthy to note that the pH of the buffer system depended strongly on the conditions, including buffer concentration and DMSO content (Appendix A). Therefore, in this manuscript, we always refer to the acetic acid to acetate ratio rather than the pH.

Since the colour appeared different at distinct acid to potassium acetate ratios, spectra were recorded for all samples in the range of 300 nm to 700 nm (Figure 1C). This showed that the samples have two maxima, one at 404 nm and one at 570 nm. The intensities of both maxima increased with the acetic acid concentration up to approximately 1 mol L^−1^. Importantly, the wavelength of the maxima did not change. In sharp contrast, the wavelength of the minimum shifted from 490 nm in the absence of acetic acid to approximately 460 nm in the presence of acetic acid concentrations of more than 1 mol L^−1^. Also, the absorbance of the minimum decreased considerably. This explains the different colour and shows that a wavelength of 570 nm is suitable for measurement.

A high buffer concentration is desirable to neutralise acids or bases present in the sample. However, high salt concentrations may impact on colour development and cause the precipitation of some compounds. To investigate this in more detail, the concentration of acetic acid varied between 0.5 mol L^−1^ and 3 mol L^−1^ and that of potassium acetate correspondingly between 0.25 mol L^−1^ and 1.5 mol L^−1^. Only slight differences in the absorption were observed and the optimum was in the range of 0.75 mol L^−1^ to 1 mol L^−1^ potassium acetate and 1.5 mol L^−1^ to 3 mol L^−1^ acetic acid (Figure 1D).

### 2.2. Type and Amount of Organic Solvent

Ninhydrin is poorly water-soluble and thus an organic solvent must be added to the reaction. In the literature, 2-methoxyethanol, phenol, DMSO and sulfolane were used. Since the first two are toxic, they were excluded from further investigations. Thus, reactions were prepared with different amounts of DMSO and sulfolane. A major difference in reagent preparation using these two solvents was that ninhydrin and hydrindantin were perfectly soluble in DMSO while they were poorly soluble in sulfolane, making heating and incubation in an ultrasonic bath necessary.

Using DMSO, an optimum of colour development was observed at a content of 35 to 45% (*v*/*v*) organic solvent in potassium acetate buffer (Figure 2A). However, the overall impact of the solvent on dye formation was small. Similarly, the sulfolane content had little impact on the reaction and only a small increase in the absorbance was observed for higher solvent contents (Figure 2B). However, the blank also increased in the same way and thus the amount of Ruhemann’s purple formed seems to be quite independent of the sulfolane content in the range of 32 to 50% (*v*/*v*) in potassium acetate buffer. The absorption was higher for reactions with DMSO than sulfolane, which has already been reported previously [54].

Since colour development was higher in DMSO, ninhydrin and hydrindantin are readily soluble in this solvent and because of its significantly lower cost than sulfolane, DMSO was used for the further experiments.

### 2.3. Ninhydrin and Hydrindantin Concentration

A (final) ninhydrin concentration between 10 and 30 g L^−1^ has been suggested in the literature and ratios of hydrindantin to ninhydrin of 1:100 to 4:100 were used [46,54,56,57]. To investigate the optimal reaction conditions, the ninhydrin concentration varied from 3 g L^−1^ to 30 g L^−1^, while the hydrindantin to ninhydrin ratio was kept at 3.75:100 in these reactions (Figure 3A). Interestingly, the curve did not pass through the origin and at the lowest tested concentration of 3 g L^−1^ only a small amount of Ruhemann’s purple was formed. At higher concentrations, a plateau of the colour formation was reached. However, the blank increased significantly with the ninhydrin concentration and thus the corrected values (colour developed in the reaction with asparagine minus blank) were calculated as well (black lines in Figure 3). This showed that the minimum concentration of ninhydrin where the plateau of colour development was reached was 20 g L^−1^. This concentration was used for subsequent experiments.

In the next step, the ninhydrin concentration was kept at 20 g L^−1^ while the concentration of hydrindantin varied from 0.1125 to 1.125 g L^−1^, corresponding to a hydrindantin to ninhydrin ratio of 0.56:100 to 5.6:100. The highest absorption was obtained at a concentration of approximately 0.8 g L^−1^ (corresponding to a hydrindantin to ninhydrin ratio of 4:100).

To this end, the reaction conditions were optimised and the optimal composition was found to be 1.6 mol L^−1^ acetic acid, 0.8 mol L^−1^ potassium acetate, 20 g L^−1^ ninhydrin and 0.8 g L^−1^ hydrindantin in DMSO/aqueous acetate buffer 40/60 (*v*/*v*) (all values refer to the final concentration). In the following experiments a mixture of the 200 µL sample and 800 µL reagent was used. Thus, the reagent must consist of 2 mol L^−1^ acetic acid, 1 mol L^−1^ potassium acetate, 25 g L^−1^ ninhydrin and 1 g L^−1^ hydrindantin in DMSO/acetate buffer 50/50 (*v*/*v*) to establish the required composition of the final reaction. As shown during optimisation, the optima are very broad and thus the reaction can tolerate small deviations from the optimal conditions.

### 2.4. Reaction Conditions

For colour development, the reaction mixtures must be heated. In the literature, different conditions have been described. Moore and Stein heated the solution for 20 min in a boiling water bath [58], Rosen for 15 min [46] and Fisher et al. only for 5 min [47].

To identify the most suitable reaction time, mixtures containing asparagine or water as a blank and the optimised reagent were heated in a block thermostat to 100 °C and samples were taken after the indicated times (Figure 4A). The best results were obtained after 30 min, but it was recognised that the colour declined upon prolonged incubation, which makes the assay less convenient since precise timing is required. Thus, a reaction temperature of 90 °C was also assayed in the same way. Here, the best results were obtained after 45 min and a broad optimum ranging from approximately 30 min to at least 60 min was observed (Figure 4B). Therefore, incubation at 90 °C for 45 min is preferable because small deviations from the optimal time have little impact on the result.

### 2.5. Response of Different Amino Acids

We used asparagine, a hydrophilic uncharged amino acid, for the optimisation of the procedure. To investigate whether the identified conditions are also suitable for other amino acids, we repeated the procedure focusing on the most significant factors: the ratio of acetic acid to potassium acetate, the concentrations of ninhydrin and hydrindantin, and the reaction time. We tested these conditions with glutamic acid, arginine, and leucine, representatives for acidic, basic, and hydrophobic amino acids, respectively.

Regarding the acetic acid to potassium acetate ratio, arginine and leucine showed, similar to asparagine, increased colour development at a higher acid content (Appendix A). In contrast, the colour development of glutamic acid was relatively independent of the acid to salt ratio. This showed that the previously established ratio is suitable for all tested amino acids. For the ninhydrin and hydrindantin concentrations, essentially the same results as for asparagine were obtained for the tested amino acids (Appendix A). Interestingly, glutamic acid, arginine and leucine reacted quicker than asparagine (Appendix A). Nevertheless, the previously suggested incubation time of 45 min is suitable for all of these amino acids. These data indicate that the aforementioned reaction conditions are appropriate.

Since the same reaction product, Ruhemann’s purple, is obtained from all primary amines, all proteinogenic amino acids should give the same response except for proline, which does not react because of its secondary amino group, and lysine, which has two primary amino groups and should therefore respond twice as strongly.

To investigate this in more detail, the responses of all proteinogenic amino acids as well as cysteine and ammonium (in form of ammonium sulphate) were investigated using the optimised reagent and reaction conditions. The response was expressed relative to that of asparagine (Table 1). The majority of the proteinogenic amino acids gave a response highly similar to that of asparagine. As expected, proline gave only a very slight response. The response of lysine was roughly twice as high as that of asparagine. Cysteine gave a significantly lower response, which is in agreement with previous reports describing that the ninhydrin reaction proceeds differently for this amino acid [59,60].

Except for these expected deviations, the other amino acids showed a very similar response, and thus the method is suitable to estimate the level of free amino acids in samples, particularly since cysteine, proline and lysine are usually scarce among the free amino acids. However, it must be mentioned that other compounds with primary amino acids and ammonia also react with ninhydrin to Ruhemann’s purple. The reactivity of ammonia was similar to that of most amino acids (Table 1).

### 2.6. Validation of the Method

The ninhydrin reaction using the optimised reaction conditions showed a highly linear response, exceeding an absorption of 2 (Appendix A).

The limits of detection and quantification were defined as the 3 and 10-fold standard deviation (SD) of the blank and were determined according to Section 4.2.2.1 of the ICH guideline Q2 (R2) on the Validation of Analytical Procedures [61]. Measuring 20 independent blank reactions gave an average absorption at 570 nm of 0.2391 with a SD of 0.0070. Using the slope of the calibration curve, this corresponds to a LOD and LOQ of 0.03 mmol L^−1^ and 0.1 mmol L^−1^, respectively.

The inter-day and intra-day repeatability were assessed using tomato juice, an aqueous extract of potato tubers and, as an example of a sample rich in free amino acids, soy sauce. As indicated in Table 2, Table 3 and Table 4, the results were highly reproducible with intra-day and inter-day relative standard deviations (RSDs) in the range of 0.5% to 2.4% and 1.7% to 2.1%, respectively. Also, the regression parameters of the calibration curves were highly similar between the different days, and all Pearson correlation coefficients exceeded 0.999 (Table 5). This shows that the developed method is highly reproducible.

The accuracy of the method was determined according to Section 3.3.1.2 of the ICH guideline Q2(R2) on the Validation of Analytical Procedures [61]. Tomato juice and soy sauce samples were spiked with asparagine at two different levels. Subsequently, the amino acid concentrations were determined in the unspiked and spiked samples (Table 6). Very good recovery rates in the range of 100% to 105% were obtained, showing that the method is very accurate.

## 3. Materials and Methods

### 3.1. Chemicals and Reagents

Ninhydrin (purity > 99%) and barium acetate (purity > 99%) were purchased from Merck (Darmstadt, Germany). Hydrindantin (purity > 97%) and all proteinogenic amino acids (all with a purity of >99%) were purchased from Sigma-Aldrich (St. Louis, MO, USA). Lithium acetate, sodium acetate, potassium acetate, magnesium acetate, calcium acetate and ammonium sulphate (all with a purity of >99%) as well as acetic acid (purity > 99.5%), 2-propanol (purity > 99.8%) and DMSO (purity > 99.8%) were purchased from Carl Roth (Karlsruhe, Germany).

Ninhydrin reagent sufficient for 24 reactions was prepared by dissolving 550 mg ninhydrin and 22 mg hydrindantin in 11 mL DMSO and mixing the solution with 11 mL acetate buffer (98.1 g potassium acetate and 111 mL glacial acetic acid dissolved in water to a final volume of 500 mL). The reagent was used directly after preparation.

### 3.2. Analytical Procedure for Quantification of Total Amino Acids

An aliquot of the sample or standard of 200 µL was mixed with 800 µL reagent in 1.5 mL reaction tubes. The tubes were centrifuged for 10 s using a Biofuge Pico centrifuge (Heraeus, Hanau, Germany) to ensure that no drops of the mixture remained in the caps of the tubes. Subsequently, the tubes were incubated in a dry block thermostat (Biosan, Riga, Latvia) at 90 °C for 45 min. Next, the tubes were cooled in water and exactly 500 µL of the reaction mixture was transferred to a 3 mL cuvette; 2.5 mL of a mixture of 2-propanol/water = 1/1 (*v*/*v*) was added and the contents were mixed well. The absorption at 570 nm was measured using a Specord 200 Plus spectrophotometer (Analytik Jena, Jena, Germany). The same instrument was also used to record the UV/VIS spectra shown in Figure 1C. A step-by-step protocol is included in the Appendix A.

During the optimisation of the reaction conditions, different variations in the reagent and procedure were used. The details are stated in the figure legends.

### 3.3. Response of Different Amino Acids

Solutions containing 2.5 mmol L^−1^ of the compound to be tested (amino acids and ammonia) were prepared and aliquots of 200 µL were reacted with 800 µL of the freshly prepared ninhydrin reagent as described above. Each time, a blank and 2.5 mmol L^−1^ asparagine standard were included, and all reactions were made in quadruplicate. The absorptions (ABS) were measured at 570 nm. The response of each compound was calculated using the formula
(1)$$Response\;in\;\%=\frac{{ABS}_{compound}-{ABS}_{blank}}{{ABS}_{asparagine}-{ABS}_{blank}}\times 100$$

## 4. Conclusions

The ninhydrin reaction has been frequently used for the determination of the total content of free amino acids in foods, plants, and other types of samples. Recently, mostly acetic acid/lithium acetate buffers at a ratio of 0–0.96/1 (mol/mol) have been used. Surprisingly, we found that a significantly higher acetic acid to acetate ratio of 2/1 gives better results with respect to the signal intensity and a more uniform dye production from different amino acids. In addition, although an acetic acid/lithium acetate buffer system was used in most previous protocols, we found that lithium, sodium, potassium and magnesium acetate are equally suitable. Due to its superior solubility in water, potassium acetate was used for the preparation of the buffer. DMSO was found to be the most suitable organic solvent because the colour development was most intense and ninhydrin and hydrindantin were readily soluble in this reagent, simplifying reagent preparation. Concentrations of 20 g L^−1^ and 0.8 g L^−1^ were found to be optimal for ninhydrin and hydrindantin, respectively. In contrast to previous publications, where samples were incubated at 100 °C for 5 to 20 min, we found that 90 °C for 45 min is more suitable because the obtained dye is more stable at that temperature. The optimised procedure was perfectly linear and highly sensitive, with an LOD and LOQ of 0.03 and 0.1 mmol L^−1^, respectively. Furthermore, the developed method was highly reproducible and accurate as indicated by the intra-day and inter-day RSDs in the range of 0.5% to 2.4% and 1.7% to 2.1%, as well as recovery rates for spiked samples in the range of 100–105%.

## Figures and Tables

**Figure 1 molecules-29-03262-f001:**
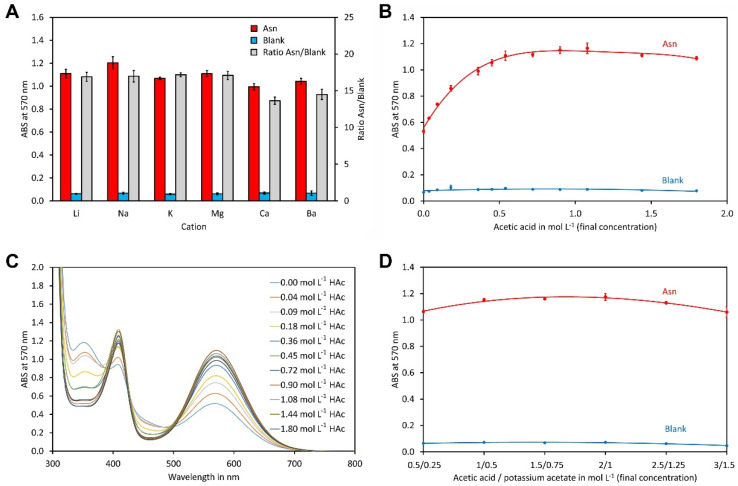
Optimisation of the buffer system. (**A**) Impact of the cation on the ninhydrin reaction. The buffer system consisted of 1 mol L^−1^ acetic acid and 0.5 mol L^−1^ acetate as salt with the indicated counter cation. (**B**) Impact of the acetic acid to potassium acetate ratio on the ninhydrin reaction. A concentration of 0.5 mol L^−1^ potassium acetate was used in all reactions while the indicated concentration of acetic acid was set using the addition of glacial acetic acid. (**C**) UV spectra of the reactions from (**B**). (**D**) Impact of the buffer concentration. All reactions contained 10 g L^−1^ ninhydrin and 0.375 g L^−1^ hydrindantin in DMSO/potassium acetate buffer 37.5/62.5 (*v*/*v*). Reactions with asparagine contained 0.5 mmol L^−1^ of the amino acid. All concentrations refer to the final concentrations in the reaction. The reactions were heated to 100 °C for 15 min. The data points and error bars represent the averages and standard deviations of four independent reactions, respectively.

**Figure 2 molecules-29-03262-f002:**
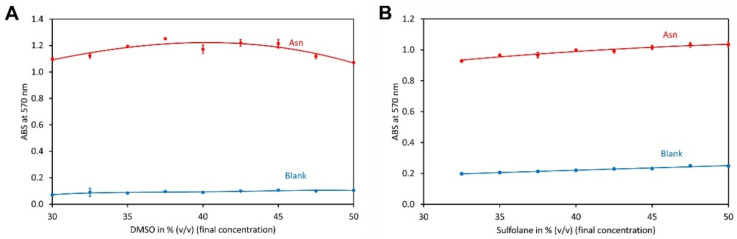
Optimisation of the organic solvent. (**A**) Impact of the DMSO content on the ninhydrin reaction. The reaction contained 1 mol L^−1^ acetic acid, 0.5 mol L^−1^ potassium acetate, 10 g L^−1^ ninhydrin 0.375 g L^−1^ hydrindantin, 0.5 mmol L^−1^ asparagine and the indicated amount of DMSO in aqueous potassium acetate buffer. All concentrations refer to the final concentrations in the reaction. The reactions were heated to 100 °C for 15 min. (**B**) Same as (**A**), but sulfolane was used as the organic solvent. The data points and error bars represent the averages and standard deviations of four independent reactions, respectively.

**Figure 3 molecules-29-03262-f003:**
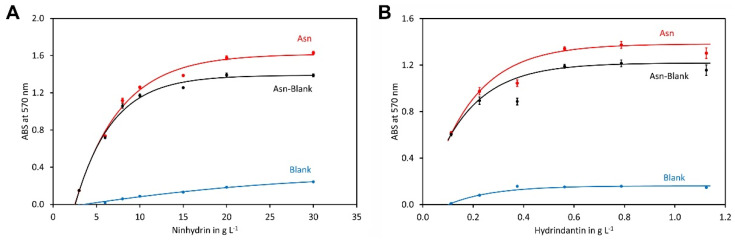
Optimisation of the ninhydrin and hydrindantin concentration. (**A**) Ninhydrin was used at the indicated concentrations while the ratio of hydrindantin to ninhydrin was kept at 3.75:100. The other compounds were used at the optimised contents and the reaction mixtures were heated to 100 °C for 15 min. (**B**) Hydrindantin concentrations were varied in the indicated range while the ninhydrin concentration was kept at 20 g L^−1^. The other conditions were the same as in (**A**). The data points and error bars represent the averages and standard deviations of four independent reactions, respectively.

**Figure 4 molecules-29-03262-f004:**
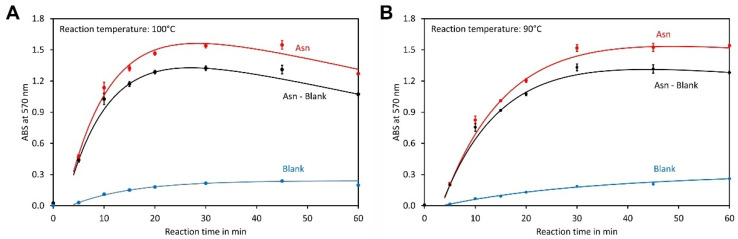
Optimisation of the reaction conditions. (**A**) Reaction at 100 °C and (**B**) 90 °C using the optimised reagent composition. The final asparagine concentration was 0.5 mmol L^−1^ in both assays. For each time point, 4 independent reactions, each with a total volume of 1000 µL, were made. The data points and error bars represent the averages and standard deviations of these reactions, respectively.

**Table 1 molecules-29-03262-t001:** Percent absorption of different amino acids based on asparagine = 100%.

Amino Acid	Response in % Based on Asn Average ± SD
Asn (Reference)	100 ± 2
Ala	96 ± 1
Arg	95 ± 2
Asp	111 ± 3
Cys	16 ± 2
Glu	117 ± 4
Gln	102 ± 2
Gly	81 ± 3
His	115 ± 1
Ile	100 ± 1
Leu	103 ± 1
Lys	212 ± 2
Met	95 ± 2
Phe	98 ± 1
Pro	6 ± 1
Ser	113 ± 1
Thr	101 ± 1
Trp	88 ± 1
Tyr	102 ± 2
Val	101 ± 1
Ammonia (NH_4_^+^)	117 ± 1

**Table 2 molecules-29-03262-t002:** Intra- and inter-day repeatability for quantification of total amino acids in tomato juice.

Experiment	Repeats	Amino Acids		
		Average mmol L^−1^	SD mmol L^−1^	RSD %
Day 1	4	32.9	0.5	1.6
Day 2	4	32.0	0.5	1.7
Day 3	4	32.7	0.2	0.5
Day 4	4	31.5	0.4	1.2
Inter-day	16	32.3	0.7	2.1

**Table 3 molecules-29-03262-t003:** Intra- and inter-day repeatability for quantification of total amino acids in potato tuber extract.

Experiment	Repeats	Amino Acids		
		Average mmol L^−1^	SD mmol L^−1^	RSD %
Day 1	4	54.30	1.28	2.4
Day 2	4	52.84	1.05	2.0
Day 3	4	53.83	0.85	1.6
Day 4	4	54.14	0.76	1.4
Inter-day	16	53.78	1.07	2.0

**Table 4 molecules-29-03262-t004:** Intra- and inter-day repeatability for quantification of total amino acids in soy sauce.

Experiment	Repeats	Amino Acids		
		Average mmol L^−1^	SD mmol L^−1^	RSD %
Day 1	4	558	4	0.7
Day 2	4	546	10	1.9
Day 3	4	560	4	0.7
Day 4	4	547	8	1.5
Inter-day	16	553	9	1.7

**Table 5 molecules-29-03262-t005:** Calibration curve equations and Pearson correlation coefficients.

Experiment	Equation	Pearson Correlation Coefficient
Day 1	0.6676x + 0.2021	0.9994
Day 2	0.6686x + 0.2175	0.9996
Day 3	0.6690x + 0.2018	0.9994
Day 4	0.6821x + 0.2027	0.9998

**Table 6 molecules-29-03262-t006:** Accuracy of the method.

Sample	Spiking Level				Found	Recovery
	mmol L^−1^	Average mmol L^−1^	SD mmol L^−1^	RSD %	mmol L^−1^	%
Tomato juice	-	29.8	0.4	1.2	-	-
	10	40.1	0.4	0.9	10.3	103
	20	50.8	0.5	1.0	21.0	105
Soy sauce	-	573	31	5.5	-	-
	500	1077	18	1.7	504	101
	1000	1568	28	1.8	996	100

## Data Availability

Raw data underlying this article are available upon request to the corresponding author.

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
