# Peer review of "The Ninhydrin Reaction Revisited: Optimisation and Application for Quantification of Free Amino Acids"

_molecules, 2024, doi:10.3390/molecules29143262_

Round 1

Reviewer 1 Report (Previous Reviewer 2)

Comments and Suggestions for Authors

The article has been partially edited since the previous posting and it will certainly interest the reader. However, the point that caused me the most doubt in the article remained without any change or comment. The comment is as follows:

I strongly recommend reworking the study of nynhydrin concentration on the resulting signal (Figure 3). If the resulting absorbances are almost 2, this means that only 1-2% of the light has passed through the cuvette. That increases the level of measurement inaccuracy. Considering that the observed effect is similar in magnitude to the measurement inaccuracy, this would benefit the reliability of the results. 

Author Response

Comment 1:

The article has been partially edited since the previous posting and it will certainly interest the reader. However, the point that caused me the most doubt in the article remained without any change or comment. The comment is as follows:

I strongly recommend reworking the study of nynhydrin concentration on the resulting signal (Figure 3). If the resulting absorbances are almost 2, this means that only 1-2% of the light has passed through the cuvette. That increases the level of measurement inaccuracy. Considering that the observed effect is similar in magnitude to the measurement inaccuracy, this would benefit the reliability of the results. 

Response 1:

Many thanks for this comment. We agree that absorbance measurements may deviate from the linearity and might be less precise at high levels. Thus, we tested the linearity of the measurement and found that it is perfectly linear up to an absorbance of 2.5 and nearly linear up to an absorbance of 3 (see Supplementary Figure S5). All measurements recorded during the optimisation procedure were clearly below 2 and thus in the perfectly linear range. Therefore, we are convinced that the data shown in the figures are reliable.

Reviewer 2 Report (Previous Reviewer 4)

Comments and Suggestions for Authors

The authors have addressed all of my comments, significantly improving their manuscript. However, the references must be updated  before their manuscript publication. 

Author Response

Comment 1:

The authors have addressed all of my comments, significantly improving their manuscript. However, the references must be updated before their manuscript publication. 

Response 1:

Many thanks for the positive evaluation! We have now completely revised the introduction and updated the references. We have reduced the number of historical references significantly but kept a few of references to older, but highly important articles. In addition, a significant number of references for recent articles have been included for the discussion of recent findings.

This manuscript is a resubmission of an earlier submission. The following is a list of the peer review reports and author responses from that submission.

Round 1

Reviewer 1 Report

Comments and Suggestions for Authors

The proposed manuscript, The Ninhydrin Reaction Revisited: Optimisation and Application for Quantification of Free Amino Acids, proposed by Stauß et al. describes accurately the use of ninhydrin in the amino acids quantification. As the authors describe, the are many methods in the frame of this work and, despite the changed conditions, the described optimization of the method does not satisfy widely the novelty. Nevertheless, the method may be extremely useful for an easy and routine quantification. In the frame of the key principles of analytical chemistry, I would like to point out the importance of standard deviation and its influence in the precision concept. It is very important to correctly express the numbers. Having done these and the following corrections, I recommend this manuscript for publication in Molecules.

1.      Abstract – row 19: the solvent, 40% (v/v) is constituted by DMSO. Which is the other solvent? Please specify it.

2.      Caption S1 – “all reaction contained 37.5% (v/v) DMSO. Which is the other solvent? Please specify it.

3.      2.2. Type and amount of organic solvent – row 205: “Using DMSO ….. 45% (v/v) organic solvent” Which is the other solvent? Please specify it.

4.      2.2. Type and amount of organic solvent – row 210: the sulfolane content in 210 the range of 32 to 50% (v/v). Which is the other solvent? Please specify it.

5.      Row 242, 245

6.      Table 1. The average and SD values must be written according to first cifra affected by the error, otherwise the SD meaning will be lose. 100.0 ± 2.3 must becomes100 ± 2; 95.4 ± 2.3 becomes 95 ± 2; 111.1 ± 2.5 must become 11 ± 3 etc.

7.      Tables 2, 3, 4 and 5. The same problem of Table 1. Please adjust values.

Reviewer 2 Report

Comments and Suggestions for Authors

The present paper deals with a textbook reaction for the determination of amino acids by ninhydrin derivatization. The authors examine the procedure in terms of ninhydrin concentration, choice of solvent, pH and type of cation in the buffer, etc. While the conclusion does not raise any major objections to the established procedure, such works are still very valuable.

I have only a few comments on the work:

1. The passage in the introduction concerning the occurrence of amino acids in food is not essential for this article and it would be good to shorten it.

2. The choice of asparagine as a test AMK is logical. I miss the optimization of reaction conditions for at least one acidic and one basic AMK. 

3. When testing the buffer composition, a molar ratio of 1/2 was chosen. I assume this is the ratio of acetate to acetic acid. Comparisons between monovalent and divalent acetate salts can be misleading. Have you taken this into consideration?

4. When testing the effect of the acid-to-salt ratio, the change in pH is likely to be the main influence. Please provide a graph of the dependence of absorbance on the pH of the solution in the Supporting materials.

5. Complete how large a volume of solution was tested for reaction temperature. Was the sample for each temperature prepared separately or were the samples taken continuously? Have you thought about whether the volume of the sample affects the reaction?

6. I strongly recommend reworking the study of nynhydrin concentration on the resulting signal (Figure 3). If the resulting absorbances are almost 2, this means that only 1-2% of the light has passed through the cuvette. That increases the level of measurement inaccuracy. Considering that the observed effect is similar in magnitude to the measurement inaccuracy, this would benefit the reliability of the results. 

Reviewer 3 Report

Comments and Suggestions for Authors

The work can not be accepted as long as you did not use chemometric experimental design models for your optimization. You have multi factors with multi levels that affect the optimum conditions. You did not care for factor-factor interactions that may lead to synergism or antagonism. You did not care for non linear responses of some factors. Accordingly the whole study must be repeated using a suitable experimental design model e.g. fractional factorial, Plackett Burman or multi level multi factor designs..etc first to find the significant factors that truly affect the response, then again to find the optimum conditions. You have to run the study again for more reliable results that can be published. This is example of a book that can enlighten you with what to do: Chemometrics: Data Driven Extraction for Science, Richard Brereton

Comments on the Quality of English Language

No problem with English

Reviewer 4 Report

Comments and Suggestions for Authors

This research enhances the ninhydrin reaction for analyzing free amino acids in diverse samples like tomato juice, potato extract, and soy sauce. Notable findings include the effectiveness of a slightly more acidic pH (around 4.3) and different acetate salts in improving results. Altering incubation conditions to 90°C for 45 minutes also enhanced dye stability.

However, the manuscript's structure needs reordering for better coherence. Additionally, the analytical validation appears incomplete and requires further details, especially regarding accuracy. There's concern about the novelty of the research for article publication, suggesting either expanding the research or considering it for a short communication.

Specific recommendations:

  • a) Replace "flavors" with "compounds" in line 28 and 36.
  • b) Emphasize the novelty in the conclusion or the last part of the introduction.
  • c) Restructure the manuscript by moving the materials and methods section before results and discussion and ensuring proper numbering.
  • d) Include CAS numbers and purity details of chemicals in the chemical and reagents section.
  • e) Suggest authors follow validation guidelines (e.g., FDA, Eurachem, ISO norms) to complete the analytical validation of the method, specifically addressing accuracy.
Comments on the Quality of English Language

I've reviewed the manuscript and believe that enhancing its readability is essential. To achieve this, I suggest eliminating unnecessary sentences and striving for greater conciseness throughout the document. Streamlining the content will greatly improve its clarity and accessibility to readers.